# Neutrophil Activity and Extracellular Matrix Degradation: Drivers of Lung Tissue Destruction in Fatal COVID-19 Cases and Implications for Long COVID

**DOI:** 10.3390/biom14020236

**Published:** 2024-02-17

**Authors:** Teluguakula Narasaraju, Indira Neeli, Sheila L. Criswell, Amita Krishnappa, Wenzhao Meng, Vasuki Silva, Galyna Bila, Volodymyr Vovk, Zolotukhin Serhiy, Gary L. Bowlin, Nuala Meyer, Eline T. Luning Prak, Marko Radic, Rostyslav Bilyy

**Affiliations:** 1Department of Microbiology, Immunology and Biochemistry, University of Tennessee Health Science Center, Memphis, TN 38163, USA; ntelugua@uthsc.edu or narasa@acu.ac.in (T.N.); ineeli@uthsc.edu (I.N.); jsilva8@uthsc.edu (V.S.); 2Department of Microbiology, Adichunchanagiri Institute of Medical Sciences, Center for Research and Innovation, Adichunchanagiri University, Mandya 571448, India; 3Department of Diagnostic and Health Sciences, University of Tennessee Health Science Center, Memphis, TN 38163, USA; scriswel@uthsc.edu; 4Department of Pathology, Adichunchanagiri Institute of Medical Sciences, Adichunchanagiri University, Mandya 571448, India; dramitay@gmail.com; 5Department of Pathology and Laboratory Medicine, Perelman School of Medicine, University of Pennsylvania, Philadelphia, PA 19104, USA; wmeng@pennmedicine.upenn.edu (W.M.); luning@mail.med.upenn.edu (E.T.L.P.); 6Department of Histology, Cytology, Histology & Embryology, Danylo Halytsky Lviv National Medical University, 79010 Lviv, Ukraine; halyna.bila@gmail.com (G.B.); r.bilyy@gmail.com (R.B.); 7Department of Pathological Anatomy and Forensic Medicine, Danylo Halytsky Lviv National Medical University, 79010 Lviv, Ukraine; vovkvi2015@gmail.com; 8Lviv Regional Pathological Anatomy Office, CU ENT (Pulmonology Lviv Regional Diagnostic Center), 79000 Lviv, Ukraine; zlthn@i.ua; 9Department of Biomedical Engineering, University of Memphis, Memphis, TN 38152, USA; glbowlin@memphis.edu; 10Institute for Immunology, Perelman School of Medicine, University of Pennsylvania, Philadelphia, PA 19104, USA; nuala.meyer@pennmedicine.upenn.edu; 11Pulmonary, Allergy, and Critical Care Medicine and Center for Translational Lung Biology, Perelman School of Medicine, University of Pennsylvania, Philadelphia, PA 19104, USA

**Keywords:** SARS-CoV-2, neutrophil elastase, elastolysis, fibrosis, collagen, myeloperoxidase, peptidyl arginine deiminase, neutrophil extracellular traps, long COVID

## Abstract

Pulmonary fibrosis, severe alveolitis, and the inability to restore alveolar epithelial architecture are primary causes of respiratory failure in fatal COVID-19 cases. However, the factors contributing to abnormal fibrosis in critically ill COVID-19 patients remain unclear. This study analyzed the histopathology of lung specimens from eight COVID-19 and six non-COVID-19 postmortems. We assessed the distribution and changes in extracellular matrix (ECM) proteins, including elastin and collagen, in lung alveoli through morphometric analyses. Our findings reveal the significant degradation of elastin fibers along the thin alveolar walls of the lung parenchyma, a process that precedes the onset of interstitial collagen deposition and widespread intra-alveolar fibrosis. Lungs with collapsed alveoli and organized fibrotic regions showed extensive fragmentation of elastin fibers, accompanied by alveolar epithelial cell death. Immunoblotting of lung autopsy tissue extracts confirmed elastin degradation. Importantly, we found that the loss of elastin was strongly correlated with the induction of neutrophil elastase (NE), a potent protease that degrades ECM. This study affirms the critical role of neutrophils and neutrophil enzymes in the pathogenesis of COVID-19. Consistently, we observed increased staining for peptidyl arginine deiminase, a marker for neutrophil extracellular trap release, and myeloperoxidase, an enzyme-generating reactive oxygen radical, indicating active neutrophil involvement in lung pathology. These findings place neutrophils and elastin degradation at the center of impaired alveolar function and argue that elastolysis and alveolitis trigger abnormal ECM repair and fibrosis in fatal COVID-19 cases. Importantly, this study has implications for severe COVID-19 complications, including long COVID and other chronic inflammatory and fibrotic disorders.

## 1. Introduction

The COVID-19 pandemic demonstrated devastating outcomes of respiratory viral infections. Although most infected individuals recover with relatively mild symptoms, many experience impaired lung function, and some require hospitalization, with as many as 1% of infected patients succumbing to the disease caused by the coronavirus SARS-CoV-2. To gain greater insight into the course of fatal COVID-19, lung autopsies from COVID-19 patients have been examined by histopathology. The disease reveals a complexity of pathologic features, including acute diffuse alveolar damage (DAD) and organized pneumonia and fibrosis [1,2,3,4,5,6]. Growing evidence indicates that during disease progression, fibrotic abnormalities become more predominant, especially in fatal COVID-19 [7,8,9,10]. Similarly, post-acute phase sequelae of COVID-19 chest computed tomography (CT) scans reveal a variable degree of fibrotic abnormalities in some recovered patients [11]. In fact, pathologic lesions of fibrosis were also noted earlier in fatal avian influenza, 2003-SARS-CoV, and Middle East respiratory syndrome coronavirus infections [12,13,14,15]. However, etiologic factors that drive virus-inflicted pulmonary fibrosis remain largely unexplored.

Pulmonary fibrosis develops in a complex interplay between cell death, inflammation, and abnormal ECM remodeling that ultimately leads to alveolar architectural disorganization, irreversible lung dysfunction, and death [16]. Recent studies associate alveolar type II pneumocyte (AT-II) epithelial injury and endoplasmic reticulum (ER)-stress-induced impairment of AT-II cell regeneration with SARS-CoV-2-inflicted fibrotic abnormalities and reveal common features with idiopathic pulmonary fibrosis (IPF) [17]. The hypothesis of elastin degradation in COVID-19 has been proposed previously [18,19]. However, the lung histopathologic or morphologic evaluation of elastin degradation and its consequences on lung abnormalities in COVID-19 patients are still unclear. Therefore, factors that drive the disorganization of ECM proteins and abnormal ECM remodeling toward fibrotic changes in acute SARS-CoV-2 infection deserve greater attention.

Elastin and collagen are major structural ECM proteins in the alveoli [20]. Elastin fibers consist of tropoelastin polymers, which are cross-linked by lysyl oxidase-mediated conversion of four lysine side chains into desmosine [21,22]. Elastin fibers align along the walls of the alveoli by association with fibrillin-1 and facilitate the expansion and recoiling of alveoli during normal breathing [23]. Unlike other ECM proteins, elastin has a half-life of 70–80 years, as the synthesis of elastin fibers occurs during the postnatal lung alveolarization and continues during the first 10 years of life in humans [24]. The regeneration/replacement of damaged elastin is inefficient, and thus damaged elastin is often replaced by collagen [25,26]. Elastin is a preferential substrate for neutrophil elastase (NE), a serine proteinase produced by activated neutrophils [27]. NE-mediated elastolysis releases elastin degradative products (EDP), potent mediators of inflammation and interstitial fibrosis [28]. Indeed, persistent neutrophil accumulation and the release of neutrophil extracellular traps (NETs) carrying NE and myeloperoxidase in the lower respiratory tracts of severely ill COVID-19 patients are associated with acute alveolar injury [29,30,31]. The cleavage of elastin is linked to the pathophysiology in injury-associated fibrosis [32], IPF [33], and chronic obstructive pulmonary disease (COPD) [34]. The activity of NE is regulated by an inhibitor called alpha-1 anti-antitrypsin (A1AT), which complexes with NE and blocks its activity [35].

In this study, eight lung autopsy samples from COVID-19 patients and six non-COVID-19 lungs were compared morphometrically. Micro-architecture with progressive elastin cleavage and collagen deposition in injured lungs correlated with lung pathophysiology, neutrophil mediated inflammation, and fibrosis. These observations reveal extensive degradation of the elastin fibers in the alveolar interstitium that accompany severe epithelial damage and massive neutrophil activation. Further, the loss of elastin and increased collagen deposits marked lesions with interstitial collagenous fibrosis and organized parenchymal fibrosis. Elastin degradation was also observed in areas of alveolar collapse, indicating that elastolysis and alveolitis contribute to abnormal ECM repair and fibrosis and lead to serious outcomes in COVID-19. As a proxy for elastase, NE-A1AT complexes in the plasma of a second group of hospitalized COVID-19 patients were quantitated. Plasma samples from hospitalized patients had approximately 30-fold higher NE-A1AT complexes than plasma from healthy donors, suggesting that such complexes may provide an indication of host tissue damage due to inflammation in respiratory lung infections.

## 2. Materials and Methods

### 2.1. Materials

Citrulline (C7629), 2,3-Butanedione monoxime (B0753), Antipyrine (A5882), Iron (III) Chloride (157740), A1AT, Bovine serum albumin (A9418), and PNPP (S0942) were purchased from Sigma-Aldrich, St. Louis, MO, USA. Antibodies including anti-NE (MAB9167, Novus Biologicals, Centennial, CO, USA), anti-A1AT (Thermo Fisher Scientific, Waltham, MA, USA), anti-peptidylarginine deiminase 4 (PAD4, ab128086, Abcam, Waltham, MA, USA), anti-MPO (14569P, Cell-signaling, Danvers, MA, USA), anti-thyroid transcription factor-1 (TTF-1, SPT24, Biocare Medical, Pacheco, CA, USA), anti-elastin (MA5-41583, Thermo Fisher Scientific, Waltham, MA, USA), and anti-histone H3 pan antibodies (07-690, Upstate, Schenectady, NY, USA) were used in this study. Soluble elastin was from Elastin Products Company, Owensville, MO, USA. Secondary antibodies for Western blot were purchased from Li-Cor Biosciences, Lincoln, NE, USA, and anti-Rabbit IgG-AP (Southern Biotech, Birmingham, AL, USA).

### 2.2. Collection of the Lung Autopsy Samples, Plasma, and Ethical Committee Approvals

Lung autopsy samples from eight patients who died of COVID-19 and six patients who died of non-COVID-19-related deaths were examined. This study was conducted in compliance with approval by the ethical committee of Danylo Halytsky Lviv National Medical University, Ukraine (Protocol Numbers: 20180226/2; 20211122/9). Patients who were hospitalized with positive COVID-19 test, with pneumonia, and with at least one complicating factor such as diabetes mellitus or hypertension and who died in the hospital after therapy including oxygen support, constituted the COVID-19 group. Non-COVID-19 samples were from hospitalized patients, who had no positive COVID-19 tests at the time of death and were characterized by oxygen saturation > 95%.

Plasma samples from hospitalized COVID-19 patients *(n*  =  62) were obtained following a protocol approved by the institutional review board (IRB# 808542), University of Pennsylvania, between April and June 2020 [36]; anonymous discarded plasma samples (*n* = 15; gift of Donald Siegel, University of Pennsylvania) [37] and additional control samples (*n* = 11) were obtained following an IRB-approved protocol from the University of Tennessee Health Science Center.

### 2.3. Histopathology, Immunohistochemistry, Morphometry, Western Blotting, Colorimetry, and ELISA

The detailed lung histopathology analysis, immunofluorescence analysis, morphometric and quantitative analysis of elastin fibers, morphometric analysis of collagen, Western blotting, determination of total citrullinated proteins, and ELISA are described in the Appendix A (see below for relevant citations).

### 2.4. Statistical Analyses

Data are expressed as the mean group value ± standard error mean (SEM). Analyses of the lung autopsies from COVID-19-positive and COVID-19-negative patients for measuring elastin fibers and collagen were performed using Image J software (version 1.54h) (National Institute of Health, April 2019; ImageJ with 64-bit Java 1.8.0_172; website: https://imagej.nih.gov/ij/, as accessed on 10 December 2023, for performing densitometry measurements. NIH). Statistical differences between the two groups were determined by Student’s *t*-test with a two-tailed comparison. A *p*-value of <0.05 was considered statistically significant.

## 3. Results

### 3.1. Characteristics of Tissue Donors

The characteristics of the subjects with fatal COVID-19 are described in Table 1. Postmortem lung autopsy samples were collected at Lviv Regional Pathological Anatomy Office, CU ENT, Pulmonology Lviv Regional Diagnostic Center, from patients who were COVID-19-positive by RT-PCR and presented with bilateral pneumonia during hospital admission (Table 1). The mean age of the subjects was 60.3 years with a range of 30 to 80 years (four female and four male patients).

Patients were admitted into the hospital on average 7.1 days from the onset of symptoms and seven of eight subjects were in the hospital for more than 10 days (range: 10 to 21 days). One patient died on day 5 after hospitalization. Prior to death, the patients had an increased percentage of blood neutrophils (non-COVID-19 patients: 57.3% vs. COVID-19 patients: 77.2%; *p* value < 0.03) and decreased lymphocytes (non-COVID-19 patients: 36% vs. COVID-19 patients: 19.1%, *p* value < 0.09). Erythrocyte sedimentation rate (ESR), a plasma marker for inflammation, was elevated in COVID-19 patients compared to non-COVID-19 patients (51.7 ± 10.7 vs. 17 ± 9.8; *p* value < 0.0004). Of the six non-COVID-19 lung autopsy samples, clinical background data were available for three patients (Table 1), who were also attended at Lviv Hospital and whose tissues were similarly prepared. Three lung samples were from patients with lung cancer but without any other known health issues, which were resected during their treatment in Memphis.

### 3.2. Histopathological Manifestations of Fibrotic Abnormalities in COVID-19

Pathologic changes including diffuse alveolar damage (DAD), and fibrosing organizing pneumonia, were evaluated in COVID-19 and non-COVID-19 lung autopsy samples (Figure 1A,B and Table 2). DAD was found in all COVID-19 patients with evidence of pervasive alveolitis (Appendix A) and sloughing of the bronchiolar epithelium (Appendix A). In a subset of patients (Table 2), the formation of a hyaline membrane (Appendix A) and intra-alveolar edema were detected (Appendix A). Consistently but to a variable extent, lung fibrosis was present in all COVID-19 patients. Mild to moderate interstitial fibrosis was present in 3/8 cases, whereas intra-alveolar fibrosis was noted in all except two cases (Figure 1D, Table 2). The onset of fibrotic changes appeared focally in the alveolar interstitium with increased deposition of collagenous fibers (Figure 1C). In most cases (5/8), organizing fibrosis extended into the surrounding parenchyma, thus causing a disorganization of alveolar architecture and loss of air spaces (Figure 1E). Fibroblast proliferation was observed, with instances of Masson bodies in alveolar and bronchiolar regions (Figure 1F). The formation of bronchiolitis obliterans organizing pneumonia (BOOP), also known as cryptogenic organizing pneumonia, was noted in one case (Figure 1G). The squamous cells formed distinct nodules resembling squamous morules (Figure 1H). Fibrotic changes also appeared in the perivascular regions with pulmonary blood vessels surrounded by loosely organized proliferating fibroblasts (Figure 1I). Vascular thrombi in the small vessels were observed in 6/8 cases (Appendix A).

### 3.3. Evidence for Elastolytic Activity in COVID-19 Patients

Recent studies identify alveolar epithelial senescence and impaired regeneration as inducing factors of fibrosis in COVID-19 [17]. However, the changes in the ECM that trigger fibrosis are unknown. The present studies revealed widespread alveolar elastolysis in COVID-19. In control lungs, elastin formed slender fibers aligned with the alveolar walls, often arranged in bundles of similar length and branching at intersections between alveoli (Figure 2A). Elastin fiberswere especially abundant near the tips of the alveolar septa. For normal lung function, elastin fibers that align with alveolar walls are essential in providing mechanical support and elastic recoil to the air spaces during respiration. In COVID-19 lungs, the elastin fibers exhibited extensive disintegration (Figure 2B). The degradation of elastin was most notable along the walls of alveoli that displayed extensive necrosis of the alveolar epithelium. Further, fragmented elastin resulted in shorter, isolated fibers that separated from the elastin bundles in the alveolar interstitial lining. The individual elastin fibers often lost their alignment within the alveolar walls.

Quantitative measurements confirmed that the lengths of elastin fibers were significantly reduced in lungs of COVID-19 patients (Figure 2C). In parallel, elastin fibers in the perivascular regions also displayed extensive dispersal in COVID-19 patients (Figure 2D,E). The loss of elastin along the blood vessels was associated with severe endothelial necrosis and distortion of the blood vessel structures. Thus, the loss of lungcapacity was compounded by the lack of elasticity in blood vessels necessary for gas exchange. In support of these findings, the Western blot analysis of elastin from protein extracts of formalin-fixed paraffin-embedded (FFPE) COVID-19 lungs identified a dramatic reduction in the tropoelastin band at 70 kDa, whereas lung extracts from COVID-19-negative patients contained clearly detectable tropoelastin (Figure 2F). The densitometry analysis (Figure 2F) revealed a significant reduction in tropoelastin concentrations in COVID-19 lungs, supporting the morphometric analysis of elastin degradation in the COVID-19 lungs.

### 3.4. Collagen Accumulates and Replaces Elastin in the Lung Parenchyma of COVID-19 Lungs

Unlike the collagen staining that appeared mainly in the alveolar walls of non-COVID-19 patient lungs, dense accumulations of collagen were observed in the alveolar interstitium in COVID-19 patients, accompanied by the thickening of the alveolar septa (Figure 3A,B). We assessed collagen content within the alveolar regions of COVID-19 and control lungs by measuring Trichrome-reactive areas, and observed a significant increase in COVID-19 (Figure 3C). In the control ECM of the alveolar interstitium, elastin fibers were generally interspersed with collagen fibers that provide elasticity and mechanical strength to the alveoli. To evaluate the relations between collagen and elastin in COVID-19 patients, we examined the expression of collagen and elastin fibers in the thin-alveolar walls of COVID-19-positive and control lungs. As shown in Figure 3D,E, epithelial injury and elastin degradation in the alveolar walls of COVID-19 lungs were associated with abundant collagen deposition. The thickened walls of the alveoli showed dense interstitial collagenous fibrosis along with degradation or a complete loss of elastin (Figure 3F–H). Interestingly, the interstitial regions showing degraded elastin also displayed epithelial necrosis, suggestingthat both elastolysis and epithelial injury are preceding events in the pathologic development of interstitial fibrosis (Figure 3D,E).

A recent study identified collapse induration associated with alveolar epithelial necrosis and the denudation of the basal lamina in a COVID-19 patient [38]. Accordingly, our study found widespread alveolar collapse with hallmarks of necrotic epithelium and narrowing of the alveolar lumen. Interestingly, the collapsed alveoli also displayed an extensive degradation of elastin fibers and, dense collagen deposition in the collapsed alveolar regions (Figure 3I,J), indicating that epithelial damage and elastolysis may precede the fibrotic changes in the injured alveoli. Similarly, perivascular regions showing loosely proliferating fibroblasts exhibited thick collagen deposits and the degradation of elastin (Appendix A). The collagen/elastin ratio was increased four-fold in the lung sections of COVID-19 patients compared to control lungs (Figure 4A–E). COVID-19lung parenchyma harboring advanced fibrosis also displayed dense collagen accumulation along with a complete absence of elastin fibers, thus suggesting that extensive elastolysis occurred during the progression of fibrosis. Areas of fibrosing organizing pneumonia displayed an extensive loss of alveolar epithelium that correlated with elastin degradation in the COVID-19 lung parenchyma (Figure 4F–H).

### 3.5. Neutrophil Aggregates and Active NETosis in the COVID-19 Lungs

Although neutrophil influx and NET release were reported in COVID-19 patient lungs, their pathogenic role is not well understood. In COVID19 lungs analyzed here, massive neutrophil aggregates were observedwithin the airways and alveolar spaces that displayed widespread disintegration of alveolar architecture (Figure 5A–D), The lung neutrophils displayed strong immunostaining for granule proteins including NE, myeloperoxidase, and PAD4, which co-localized within the regions of neutrophil aggregates in the alveoli (Figure 5E). Control lungs showed reduced immunoreactivity for NE, MPO, and PAD4 (Figure 5F–H). Quantitative measurements identified an increase in total citrullines in COVID-19 lungs compared to controls (Figure 5I). Strong NE, MPO, and PAD4 staining in the extracellular space within the disintegrated alveoli is consistent with the degranulation of neutrophils in the inflammatory microenvironment.

### 3.6. Increased NE-A1AT Complexes in Plasma from COVID-19 Patients

Given the accumulation of neutrophils in COVID-19 patient lungs and the abundant evidence of NE expression and activity in the tissues, it was of interest to determine whether plasma from COVID-19 patients could provide evidence of increased NE abundance. Because NE activity quickly dissipates, due to the high levels of the A1AT protease inhibitor that binds to NE and blocks its activity [39], we considered the appealing alternative to measure concentrations of NE-A1AT complexes in plasma from sixty-nine COVID-19 patients and twenty-five non-COVID-19 controls. A sandwich ELISA revealed a significant increase in NE-A1AT complexes in plasma from hospitalized COVID-19 patients (Figure 5J).

## 4. Discussion

Although the COVID-19 pandemic appears to be subsiding after three years of global devastation, sporadic outbreaks continue to recur in different parts of the world, especially during winter seasons [40,41,42]. Mortality rates remain high in hospitalized patients with severe infections. Our examination of lungs from patients who died of COVID-19 revealed a variety of pathologic manifestations including acute DAD, impaired lung repair, and fibrosis. Notably, we observed direct evidence for neutrophil activation and inferred neutrophil contribution to lung fibrosis in COVID-19 patients who succumbed to the disease. The underlying mechanisms of how SARS-CoV-2 infection progresses to fibrotic abnormalities in injured lungs motivated the current study.

The pathophysiological development of pulmonary fibrosis involves a series of events that include massive inflammation, alveolar type II epithelial damage, subsequent surfactant deficiency, and abnormal ECM remodeling. Overall, the progression of the disease manifests in alveolar collapse, collapse induration, and fibrosis [43,44]. Epithelial loss and insufficient epithelial regeneration in alveoli trigger the expression of pro-fibrotic cytokines such as transforming growth factor beta, IL-6, and tumor necrosis factor-alpha, all known to induce fibrotic remodeling [45]. Recent observations highlight AT-II epithelial injury and ER-stress-induced arrest of AT-II cell regeneration as promoters of fibrotic changes in COVID-19 patients [17,46]. The present study investigated the role of neutrophils in the damage to the structural integrity of ECM proteins, the inverse relation between elastin and collagen, and the accumulation of fibrotic abnormalities in the lungs of fatal COVID-19 patients. Our study has implications for patients with long COVID-19-associated lung impairment, which often involves fibrotic scarring of lungs that drastically reduces lung function. Consistent with our data, Woodruff et al. [32] examined post-acute sequelae of COVID-19 and inferred from a proteomic analysis that immune-mediated fibrosis and ongoing neutrophil activity are associated with inflammation that persists for over one year after the acute phase of the SARS-CoV-2 infection. Interestingly, this extended aftermath of the infection is also characterized by the induction of newly arising autoreactivity.

The most notable result of our study was the overall degradation of elastin in the extracellular matrix of COVID-19 lungs. This degradation included the cleavage and disintegration of the elastin fibers of the alveolar wall, as well as within the critical vascular network in the lung parenchyma. Damaged lungs with pathologic manifestations of fibrosing organizing pneumonia displayed complete degradation or an absence of the elastin network. Elastin is a polymer of tropoelastin units that scaffold onto fibrillin-1 along the walls of the alveoli [21]. Structurally, elastin provides elasticity and tensile strength essential for normal stretching and contraction of the alveoli during respiration. Thus, a loss of elastin fibers potentially destabilizes alveoli, promoting their collapse. Elastolysis has been associated with inflammation-driven neutrophil recruitment in several acute and chronic clinical conditions [32,33,34].

The dysregulated neutrophil activation and overwhelming NE activity contribute to elastin degradation and to the disruption of the alveolar basement membrane, causing DAD and potentially other secondary complications. The NE-mediated release of elastin degradative products has been shown to enhance myofibroblast differentiation [47]. In the present study, an analysis of lung autopsies illustrated a dense neutrophil accumulation within the areas of DAD with overwhelming expression of NE and MPO in the airways and alveolar spaces. Further, an increase in PAD4 expression indicated active NETosis in the infected lung microenvironment of COVID-19 patients. Consistent with NETosis, increased levels of total citrullinated proteins were observed in lung autopsy samples of COVID-19 patients compared to controls.

PAD4 activation, resulting in protein deimination or “citrullination”, is reliably induced in neutrophils responding to inflammation [48] and arguably contributes to sequelae of long COVID-19 [49].

Numerous clinical reports also found increased neutrophilia, high neutrophil-lymphocyte ratios [50,51], and NETosis associated with progressive lung damage [52]. Our data revealed evidence of thrombosis in five of eight COVID-19 lung autopsies, thus suggesting a role for NETs in this critical aspect of lung pathology [53]. Plasma MPO-DNA complexes may be a particularly relevant marker for COVID-19 thrombogenesis [54]. Consistent with the central role of MPO in the lung pathology observed here, RNF128, an endogenous inhibitor of MPO, was observed to limit lung damage in a model of acute lung injury [55]. These results emphasize that the aggressiveness of invading neutrophils results in profound damage to the ECM in the lung parenchyma and indicate that a range of neutrophil activities directly conspire to extensively degrade elastin.

Strikingly, a morphometric analysis of alveolar interstitium shows that elastin degradation is associated with increased collagen deposition and resulting interstitial fibrotic changes. In support of this association, collagen/elastin ratios showed increased collagen deposition in the lung parenchyma in advanced fibrotic regions, which displayed widespread alveolar epithelial disintegration in COVID-19 patients. Thus, a combination of elastolysis with epithelial and endothelial injury disrupts the alveolar–capillary barrier; increased alveolar rigidity may cause a remodeling of the alveolar architecture, eventually inducing collagen expression and, ultimately culminating in engulfing fibrosis. These results support the view that elastolysis combined with epithelial injury precedes collagen deposition and is likely to progress into interstitial fibrosis. The disintegration of alveolar architecture compromises gas exchange at the thin epithelial–capillary network in the alveoli. Thus, our study supports the model that elastolysis and alveolitis lead to collagenous fibrotic abnormalities, ultimately contributing to respiratory dysfunction in severely ill COVID-19 patients.

These findings also establish a direct connection between pulmonary sequestration and the degranulation of neutrophils, with the destruction of the alveolar architecture. Many views of COVID-19-affected lungs show granulocytes and identify colocalized granule enzymes [18,29]. Among these, NE is the obvious candidate for the massive degradation of elastin fibers that precede fibrotic abnormalities. Clearly, NE can be placed at the scene of matrix dissolution by immuno-fluorescence. However, it is difficult to measure changes in NE more distantly, such as in plasma from affected individuals. This is because NE that is locally released is rapidly inhibited by A1AT through the formation of NE-A1AT complexes. Interestingly, an analysis of plasma samples of COVID-19 patients showed high levels of NE-A1AT complexes, indicating elevated extracellular release of NE. The formation of NE-A1AT complexes thus represents a candidate biomarker for the diagnosis of neutrophil activation, NET release, and tissue damage in COVID-19 patients. These observations need further investigation to validate the associations reported here with the most severe outcomes of COVID-19.

## 5. Conclusions

In summary, the present study provides insights into the pathologic manifestations of COVID-19 and highlights that elastolytic activity plays a key role in exacerbating pulmonary pathology in COVID-19 patients, especially in the development of pulmonary fibrosis. Dysregulated neutrophil activity and NETosis contribute to elastin degradation, ECM remodeling, and fibrotic changes in COVID-19. A better understanding of the mechanism of elastin degradation and its significance for the pathology of fibrosis will likely help to identify novel therapeutic targets and prevent pulmonary fibrosis in COVID-19 patients. Importantly, our study may help to guide efforts to reduce or ameliorate the excessive activity of neutrophils that contributes to debilitating effects of long COVID.

## Figures and Tables

**Figure 1 biomolecules-14-00236-f001:**
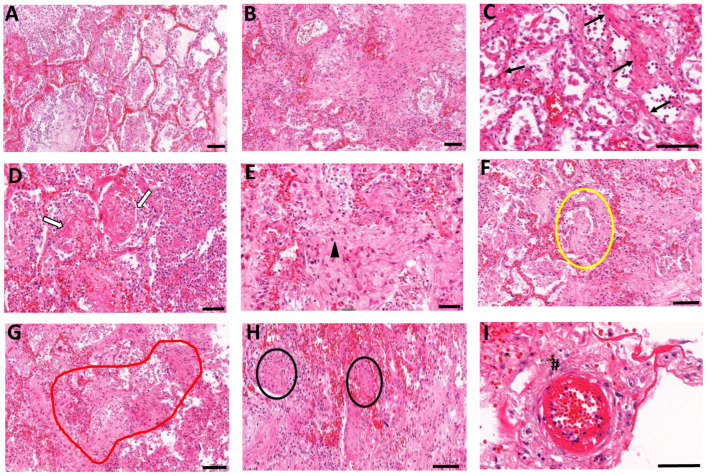
Lung histopathology analyses of COVID-19 patients. Autopsy lung samples collected from COVID-19 patients show major patterns of (**A**) widespread diffuse alveolar damage and (**B**) fibrosing organizing pneumonia. Fibrotic changes in COVID-19 patients show variable patterns including (**C**) interstitial fibrotic development (black arrows), (**D**) intra-alveolar fibrin deposition and organizing fibrosis, showing spreading of fibrosis in alveoli (white arrows), and (**E**) diffuse alveolar fibrosis within small airways (arrowhead). (**F**) Loose myxoid fibroblastic proliferation extended between the alveoli-forming Mason bodies in alveolar and bronchiolar regions (yellowoval). (**G**) Bronchiolitis obliterans organizing pneumonia (BOOP)-like areas (redoutline). (**H**) Squamous cells forming distinct nodules (blackovals). (**I**) Disrupted small blood vessels also displayed perivascular fibroblast proliferations (hash mark). Scale bars = 100 μm.

**Figure 2 biomolecules-14-00236-f002:**
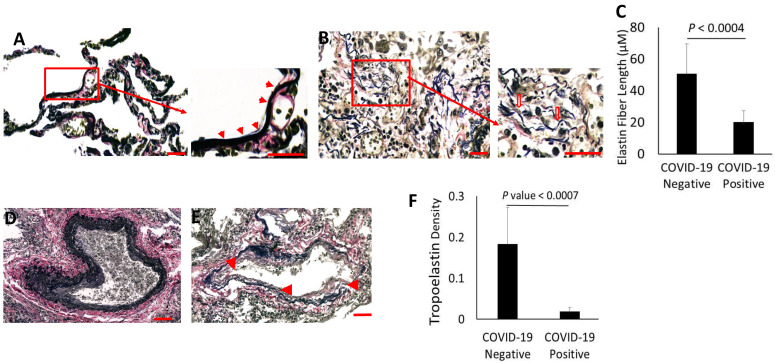
Elastin degradation in the lungs of the COVID-19 patients. The quantitative measurement of elastin fiber lengths in μm. The elastin fibers present in the alveolar walls were measured using ImageJ analysis. (**A**) COVID-19 negative patients. Elastin fibers appear as bundles in the alveolar walls (red arrows). (**B**) In COVID-19 patients, the fragmentation of elastin fibers was observed (open red arrow). (**C**) The lengths of elastin fibers in COVID-19-positive and COVID-19-negative lungs were measured from 10 randomly selected fields per sample. Elastin fiber lengths are represented as means ± SD in lung images of COVID-19 (*n* = 8) and non-COVID-19 sections (*n* = 6). (**D**,**E**) Elastolysis (red arrowhead) in the perivascular regions of the COVID-19 patients compared to COVID-19 negative patients. (**F**) The lung tissue extracts from COVID-19 negative (*n* = 4) and COVID-positive (*n* = 6) FFPE sections were probed with anti-elastin antibodies. The intensities of tropoelastin immunofluorescence were normalized against histone and plotted as relative fluorescence units (FU). Scale bar = 100 μm.

**Figure 3 biomolecules-14-00236-f003:**
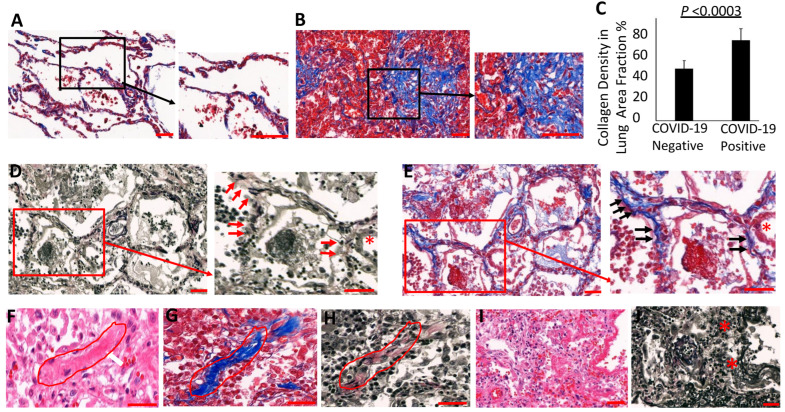
Loss of elastin is accompanied by collagenous fibrosis in the alveolar interstitium and perivascular regions. Collagen was measured in (**A**) COVID-19 negative and (**B**) COVID-19 positive lungs using Masson’s trichrome collagen-specific staining. (**C**) The density of collagen in the lung was quantified using ImageJ analysis as described in the methods and at least 10 randomly selected fields from each lung section were measured. Collagen density measurements were represented as means ± SD in lung sections from COVID-19 patients (*n* = 8) and non-COVID-19 patients (*n* = 6). (**D**) Necrotic and denuded alveolar epithelial lining displayed elastolysis (red arrows—loss of elastin in the alveolar walls; red asterisk—epithelial denudation; black arrows—collagen accumulation) and (**E**) dense collagen deposition. (**F**–**H**) Collagen replacing elastin fibers in the formation of interstitial fibrotic changes in COVID-19 patients evident by H&E, collagen deposition, and elastin degradation, respectively (redoutline). (**I**,**J**) Lung parenchyma shows alveolar collapse (red asterisks) fibrotic development and reduction in elastin fibers. Scale bar = 100 μm.

**Figure 4 biomolecules-14-00236-f004:**
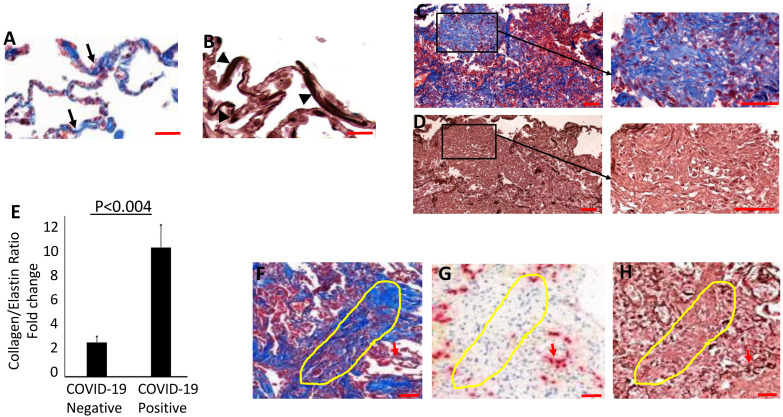
Measurement of collagen/elastin ratio. The density of collagen and elastin fibers’ distribution in the lung parenchyma was measured in COVID-19-positive and COVID-19-negative patients. Adjacent lung sections stained with Masson’s trichrome and orcein from a non-COVID-19 patient ((**A**), collagen—black arrows and (**B**), elastin—black arrowheads). COVID-19 patient lungs displayed (**C**) widespread collagen deposition and (**D**) a complete loss of elastin. (**E**) The collagen/elastin ratio was quantified using ImageJ analysis as described in the methods. (**F**–**H**) Areas of fibrosing organizing pneumonia also displayed extensive loss of alveolar epithelium stained with TTF-1 ((**G**); yellow outline), while undamaged areas showed positive staining for TTF-1 ((**G**); red arrow) and elastin ((**H**); red arrow). The values of collagen/elastin densities were represented as means ± SD in lung sections from COVID-19 patients (*n* = 8) and non-COVID-19 patients (*n* = 6). Scale bar = 100 μm.

**Figure 5 biomolecules-14-00236-f005:**
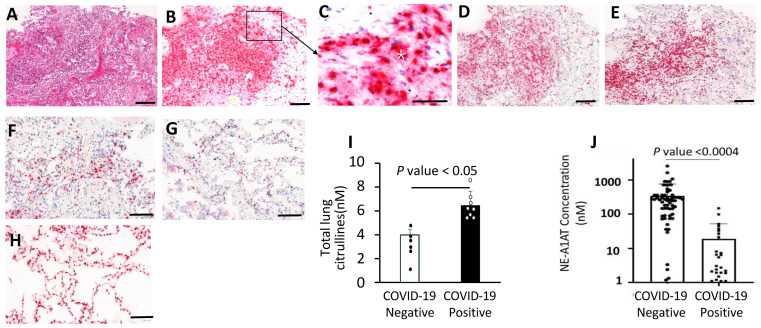
Neutrophil influx, enzyme activities, and NETosis are increased in the lung parenchyma of COVID-19patients. (**A**) H&E staining of lung tissues with massive neutrophil influx. (**B**) Neutrophils show high expression of neutrophil elastase. (**C**) Insert from panel B depicts extracellular release of NE (white asterisk) in the alveoli among numerous neutrophils. Lung sections with immunohistochemistry for MPO (**D**) and PAD4 (**E**) support neutrophil enzyme activation. (**F**–**H**) NE, MPO, and PAD4 staining in non-COVID-19 lungs. (**I**) Increased levels of total citrullinated proteins in the COVID-19 lungs (*n* = 8) compared to non-COVID-19 extracts (*n* = 6). (**J**) Evaluation of NE-A1AT complex in the plasma samples from independent cohorts of COVID-19 patients by ELISA displayed increased NE-A1AT complexes in COVID-19 patients (*n* = 69) compared to healthy individuals (*n* = 25). The data are represented as means ± SD. Scale bar = 100 μm.

**Table 1 biomolecules-14-00236-t001:** Summary of autopsy specimens.

No.	Sex	Age	Upon Hospitalization	Days in the Hospital	Complications	Disease Duration	SARS-CoV-2 PCR Positive, at Day	SpO_2_				ESR		Hb		WBC	Lymphocytes				Neutrophils				Platelets, ×10^3^/uL	
			For all—cause of death is Acute hemorrhagic tracheobronchitis, respirators distress syndrome (according to local classification).					upon hospitalization		last record		upon hospitalization	last record	g/L	Hct %	last record	upon hospitalization	upon hospitalization	last record	last record	upon hospitalization	upon hospitalization	last record	last record	upon hospitalization	last record
								with O_2_	air	with O_2_	air					×10^9^/L	×10^9^/L	%	×10^9^/L	%	×10^9^/L	%	×10^9^/L	%		
204	M	61	bilateral pneumonia COVID-19	10	other viral and bacterial pneumonia, acute respiratory syndrome, hypertensive heart disease, chronic lymphocytic leukemia, arterial hypertension, gastric ulcer	16	6	n/a	90	80	n/a	55	46	99	29.6	25.2	18.3	72.5	26.6	61.2	4.6	18.3	12.5	24	210	280
205	F	64	bilateral pneumonia COVID-19	21	tongue carcinoma, hypertension, hypothyroidism	27	6	n/a	96	97	n/a	72	49	96	23.4	4.7	0.2	14.8	0.4	8.7	1	80.8	4.2	87.4	202	361
208	M	59	bilateral pneumonia COVID-19, DM	5	hypertension, diabetes mellitus, varicose disease	8	2	n/a	86	85	n/a	45	42	135	39.1	6.5	0.76	13	0.5	8.1	4.63	79	58	89	155	155
210	F	80	bilateral pneumonia COVID-19	19	hypertension, kidney disease, COPD	26	7	98	n/a	73	55	47	24	147	44.2	14.3	0.2	n/a	0.9	6.1	11.4	84.1	13	91	179	179
211	F	61	bilateral pneumonia COVID-19	19	hypertension, diabetes mellitus, COPD	24	5	78	65	98	n/a	42	17	141	40.8	10.5	1.1	10.6	1	14.9	8.6	81.9	5.5	80.30	143	389
212	M	30	pneumonia (unilateral, R) COVID-19	17	pneumothorax (R)	26	9	97	90	78	62	53	42	100	30.1	18.6	0.6	9	0.5	2.5	11.6	95	17.4	93.8	238	324
214	M	71	pneumonia (unilateral, R) COVID-20	16	bacterial pneumonia	24	8	97	81	80	n/a	37	9	162	46	10.2	0.7	6.9	1.1	10.8	9.6	89.2	5.26	85.5	225	226
216	F	56	bilateral pneumonia COVID-19	13	hypertension, diabetes mellitus, obesity, multiple organ failure	27	14	90	n/a	70	n/a	49	29	120	36.4	6.5	0.4	7.2	0.4	6.5	5	89.1	5.7	87.7	287	421
mean		60.3		15.0		22.3	7.1	92.0	84.7	82.6	58.5	50.0	32.3	125.0	36.2	12.1	2.8	19.1	3.9	14.9	7.1	77.2	15.2	79.8	204.9	291.9
809	F	81	cecum cancer pT3N0M0G2	4	emphysema, anemia, atherosclerosis, thromboemboly of lung artery		0		98		98	28		78		4.6	4.6	32			3.5	58			242	
825	M	85	atherosclerotic aneurism of infrarenal aorta with rupture	1	hypertension, obesity, chronic bronchitis	1	0		95			9		124	45	6.8	6.2	37			6.2	62			280	
835	F	66	hypertensive microangiopathy of brain, kidney, and pancreas	21	fracture L2-L4 7 years ago		0		96		96	14	12	132	42	5.4	5.4	39			5.5	52			245	
mean		77.3		8.7		1.0	-		96.3		97.0	17.0	12.0	111.3	43.5	5.6	5.4	36.0			5.1	57.3			255.7	

Note: Three non-COVID-19 lung resections were received at UTHSC, from patients with lung cancer but no other known health issues (Neg). n/a, not applicable.

**Table 2 biomolecules-14-00236-t002:** Histopathologic signatures of the lung autopsies of COVID-19 patients.

Variables	Case 1	Case 2	Case 3	Case 4	Case 5	Case 6	Case 7	Case 8
**Alveolar changes**								
Hemorrhage	1	1	1	3	0	1	1	2
Edema	1	2	0	0	1	3	0	3
Fibrin deposition	1	0	0	3	1	3	1	1
Hyaline membrane	1	1	0	0	0	1	0	1
Exfoliation	1	1	1	2	2	1	1	1
Necrosis of epithelium	1	1	1	0	0	1	1	1
Type 2 Pneumocyte hyperplasia	3	3	1	3	1	3	3	3
Organization (fibrosis)	0	2	3 Bacterial colonies BOOP Masson bodies	3	3	1	2	0
Inflammation	0	2	3	2	3	3	0	1
Multinucleate giant cells	0	0	0	0	0	1	0	0
Others	Atypia of type II pneumocytes	Atypia of type II pneumocytes	Mitosis Reed Sternberg like cells squamous modules	Atypia of type II pneumocytes	Acute eosinophilic pneumonia	Corpora amylacea	-	-
**Interstitial changes**								
Hemorrhage	2	2	3	3	0	1	1	0
Expansion	2	2	3	3	2	1	2	1
Inflammation	2	2	3	1	1	1	1	1
Fibrosis	1	2	2	3	0	0	3	1
Fibrin deposition	1	1	1	3	0	1	1	1
Vasculitis	1	1	0	1	0	1	1	1
Thrombi	1	1	0	2	0	1	1	0
Pleura	1	0	0	0	0	0	0	0
Bronchiolitis	1	1	1	1	1	1	1	1

Alveolar changes: Hemorrhage: Absent—0, Mild—1, Moderate—2, Severe—3; Edema: Absent—0, Mild—1, Moderate—2, Severe—3; Fibrin deposition: Absent—0, Mild—1, Moderate—2, Severe—3; Hyaline membrane formation: Absent—0, Mild—1, Moderate—2, Severe—3; Type II pneumocyte hyperplasia: Absent—0, Mild—1, Moderate—2, Severe—3; Organization (fibrosis): Absent—0, Mild—1, Moderate—2, Severe—3; Inflammation: Absent—0, Mild—1, Moderate—2, Severe—3; Epithelial necrosis: Absent—0, Present—1; Exfoliation: Absent—0, Present—1; Alveolar atypia: Absent—0, Present—1; Multinucleate giant cells: Absent—0, Present—1; Bronchiolitis obliterans organizing pneumonia (BOOP). Interstitial changes: Expansion: Absent—0, Mild—1, Moderate—2, Severe—3; Inflammation: Absent—0, Mild—1, Moderate—2, Severe—3; Fibrosis: Absent—0, Mild—1, Moderate—2, Severe—3; Fibrin deposition: Absent—0, Mild—1, Moderate—2, Severe—3; Vasculitis: Absent—0, Present—1; Thrombi: Absent—0, Few vessels (1–3)—1, Many vessels (>4)—2); Pleura-Chronic inflammation: Absent—0, Present—1; Bronchiolitis: Absent—0, Present—1.

## Data Availability

The data presented in this study are available on request from the corresponding author.

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
