# Peer review of "Neutrophil Activity and Extracellular Matrix Degradation: Drivers of Lung Tissue Destruction in Fatal COVID-19 Cases and Implications for Long COVID"

_biomolecules, 2024, doi:10.3390/biom14020236_

Round 1

Reviewer 1 Report

Comments and Suggestions for Authors

This study examined lung tissue from eight COVID-19 and six non-COVID-19 postmortems to investigate changes in extracellular matrix (ECM) proteins, including elastin and collagen, within lung alveoli. The findings show significant degradation of elastin fibers in the thin alveolar walls of the lung, which happens before the deposition of interstitial collagen and widespread intra-alveolar fibrosis. Lungs with collapsed alveoli and organized fibrotic areas exhibited extensive elastin fiber fragmentation and alveolar epithelial cell death. Elastin degradation was confirmed through immunoblotting. These findings suggest that neutrophils and elastin degradation are central to impaired alveolar function, leading to abnormal ECM repair and fibrosis in fatal COVID-19 cases. Additionally, this study has implications for severe COVID-19 complications, long COVID, and other chronic inflammatory and fibrotic disorders.

The study provides useful mechanistic insights in the favor for publication.

Explain more details on the sampling procedures, the place and country of the patients or the source of lung samples

There seem to be different sampling places, clarification is needed on sampling heterogeneity

Section 2.3 must be given in more details

Provide more details of the demographic analysis of the population “the patients used in sampling”

Table 2 needs statistical analysis using the appropriate methods

Please rephrase “elastin fibers in control alveoli appeared as slender fibers aligned with the alveolar walls,………….

Author Response

Response to Reviewer 1:

The authors thank the Reviewer 1 for the careful consideration of our study and the thoughtful summary of our findings. We also thank the Reviewer for supporting the eventual publication.

  1. The Reviewer ‘s point is astute. The biological samples were obtained at the different locations and care was taken to prepare tissues the same way. Importantly, sample processing and immunochemistry were performed in parallel by the same histologist for both COVID-19 and control samples. This point is clarified on pages 4 and 5 of the revised text.
  2. The Section 2.3 was abbreviated to avoid burdening the readers with the description of accepted and standard methods. The Supplementary Methods covers specific aspects of laboratory procedures, as noted in Section 2.3.
  3. The demographics have been listed and are white Caucasians for all tissues collected at the Kyiv location.
  4. We acknowledge and appreciate the comment from the Reviewer that Table 2 needs to have a statistical analysis. However, the individual scores represent defined categories of observable phenotypes, which are applied by a pathologist in a blinded fashion. Thus, obtained numbers are not strictly linear and should not be represented on a linear scale. For that reason, a calculation of means and standard deviations is not appropriate.
  5. We thank the Reviewer for pointing out the stylistically awkward sentence and have now attempted to improve it. The amended sentence is on page 7 and is highlighted in yellow. It reads: “In control lungs, elastin formed slender fibers aligned with the alveolar walls,..”

Reviewer 2 Report

Comments and Suggestions for Authors

Narasaraju et al. performed histopathological analysis of lung tissue specimens from COVID-19 and non-COVID-19 postmortems and revealed significant degradation of elastin fibers, ECM remodeling with an increased collagen/elastin ratio, fibrotic changes, and dysregulated neutrophil activation with massive neutrophil aggregates that displayed strong immunostaining for NE, MPO and PAD4 in the lung tissue specimens from COVID-19 patients. In addition, the increased NE-A1AT complexes were detected in plasma from COVID-19 patients

Manuscript is clear, well-written and addresses the important clinical issue. However, there are some points that should be considered and adequately addressed:

- MPO activity in plasma samples should be included and/or discussed in the manuscript

- Being few samples (6-8), please show the individual points in all the graphs to allow the readers to understand how much the values vary from one another

- Figure 5 panel should also contain H&E staining as well as the immunohistochemical staining for MPO and PAD4 of lung tissue sections of COVID-19-negative patients

- Thrombus formation was obtained within interstitial changes in 5 out of 8 lung autopsies of COVID-19 patients (Table 2) The contribution of increased collagen/elastin ratio as well as NET formations in thrombogenesis should be discussed

- In Table 1 total WBC count should be included. The absolute lymphocytes count for 3 COVID-19-negative patients upon hospitalization are included, while the absolute neutrophils counts are missing

Author Response

We thank the Reviewer for the positive and constructive review of our manuscript. Detailed responses are given below.

  1. As described by Shih et al. (2008), MPO in plasma is sensitive to various aspects of sample collection, storage, handling, and buffer conditions. The authors note particular concerns, which may apply to the samples in our analyses. One concern is that assays of of MPO levels becomes less reliable if -20C storage extends for longer than 3 months. Second, an MPO assay is affected by conditions used for plasma collection and the number of freeze/thaw exposures. On the other hand, FFPE preserved samples used for morphometric analysis by immunohistochemistry is considered quite resistant to handling.
  2. We thank the Reviewer for this comment. We wish to clarify that, as described in the Supplementary Methods, the morphometric measurements were done by randomly selecting 10 different views of each slide for each sample and examine each view with Image J software to measure and determine the elastin fiber length, collagen density, and ratios between them from microscopy. Therefore, individual points may be differentially distributed across one view but will randomize by comparison to other views. Each sample provides ten independent views, which are averaged into a composite median value for each slide. The 80+ individual data points would be difficult to accurately perceive. For that reason, we maintained morphometric data as simple bar graphs, but we will provide any additional data which we have upon request.  However, we are pleased to fulfill with the Reviewer’s request and display individual data points for measurements of total citrulline content in solubilized FFPE sections (revised Figure 5) and anti-tropoelastin immunofluorescence reactivity in solubilized and recovered protein extracts (Figure 2G). Where we already presented individual data points in the originally submitted manuscript, such as the ELISA determination of A1AT-NE complexes (Figure 5G), we did not institute a change.
  1. As requested by the Reviewer, the revised Figure 5 now also displays IHC for MPO, PAD4 and NE against non-COVID lung sections.
  2. We thank the Reviewer for suggesting the additional statements in the Discussion. Now, the revised text reads as follows (shown in yellow on page 12): “Our data revealed evidence of thrombosis in 5 of 8 COVID-19 lung autopsies, thus suggesting a role for NETs in this critical aspect of lung pathogenesis [53]. Plasma MPO-DNA complexes may be a particularly relevant marker for COVID-19 thrombo-genesis [54]. Consistent with the central role of MPO in the lung pathology observed here.” These statements made it necessary to include new citations of the papers by Zuo et al. and Middleton et al. We also considered the Reviewer’s point more broadly and connected PAD4 to disease initiation and progression (new references 48 and 49). “PAD4 activation, resulting in protein deimination or “citrullination”, is reliably induced in neutrophils responding to inflammation [48] and arguably contributes to sequelae of long COVID-19 [49].”
  3. Table 1 has been revised as requested by the Reviewer, and now includes the WBC counts for all COVID-19 samples and displays the absolute neutrophil counts. We thank the Reviewer for the thorough and helpful review of the data.

Citations:

Shih J, Datwyler SA, Hsu SC, Matias MS, Pacenti DP, Lueders C, Mueller C, Danne O, Möckel M. Effect of collection tube type and preanalytical handling on myeloperoxidase concentrations. Clin Chem. 2008 Jun;54(6):1076-9. doi: 10.1373/clinchem.2007.101568. PMID: 18509013.

Zuo Y, Zuo M, Yalavarthi S, Gockman K, Madison JA, Shi H, Woodard W, Lezak SP, Lugogo NL, Knight JS, Kanthi Y. Neu-trophil extracellular traps and thrombosis in COVID-19. J Thromb Thrombolysis. 2021 Feb;51(2):446-453. doi: 10.1007/s11239-020-02324-z. Epub 2020 Nov 5. PMID: 33151461

Middleton EA, He XY, Denorme F, Campbell RA, Ng D, Salvatore SP, Mostyka M, Baxter-Stoltzfus A, Borczuk AC, Loda M, Cody MJ, Manne BK, Portier I, Harris ES, Petrey AC, Beswick EJ, Caulin AF, Iovino A, Abegglen LM, Weyrich AS, Rondina MT, Egeblad M, Schiffman JD, Yost CC. Neutrophil extracellular traps contribute to immunothrombosis in COVID-19 acute res-piratory distress syndrome. Blood. 2020 Sep 3;136(10):1169-1179. doi: 10.1182/blood.2020007008. PMID: 32597954

Round 2

Reviewer 1 Report

Comments and Suggestions for Authors

Accept